# IMPLICIT RUGOSITY REGULARIZATION VIA DATA AUGMENTATION

## ABSTRACT

Deep (neural) networks have been applied productively in a wide range of supervised and unsupervised learning tasks. Unlike classical machine learning algorithms, deep networks typically operate in the *overparameterized* regime, where the number of parameters is larger than the number of training data points. Consequently, understanding the generalization properties and the role of (explicit or implicit) regularization in these networks is of great importance. In this work, we explore how the oft-used heuristic of *data augmentation* imposes an *implicit regularization* penalty of a novel measure of the *rugosity* or "roughness" based on the tangent Hessian of the function fit to the training data.

## 1 INTRODUCTION

Deep (neural) networks are being profitably applied in a large and growing number of areas, from signal processing to computer vision and artificial intelligence. The expressive power of these networks has been demonstrated both in theory and practice (Cybenko, 1989; Barron, 1994; Telgarsky, 2015; Yarotsky, 2017; Hanin & Sellke, 2017; Daubechies et al., 2019). In fact, it has been shown that deep networks can even perfectly fit pure noise (Zhang et al., 2016). Surprisingly, highly *overparameterized* deep networks—where the number of network parameters exceeds the number of training data points—can be trained for a range of different classification and regression tasks and perform extremely well on unobserved data. Understanding why these networks generalize so well has been the subject of great interest in recent years. But, to date, classical approaches to bound the generalization error have failed to provide much insight into deep networks.

The ability of overparameterized deep networks to overfit noise while generalizing well suggests the existence of some kind of *regularization* in the learning process. Sometimes the regularization is *explicit*, as in the case of Tikhonov or $\ell_1$ regularization (Krogh & Hertz, 1992; Bishop, 1995; Han et al., 2015; Bartlett et al., 2017). By contrast, we are becoming increasingly aware that various heuristic techniques—such as dropout (Baldi & Sadowski, 2013; Wager et al., 2013; Srivastava et al., 2014), batch normalization (Ioffe & Szegedy, 2015; Salimans & Kingma, 2016; Bjorck et al., 2018), and even stochastic gradient descent (Hardt et al., 2015; Bousquet & Elisseeff, 2002; Yao et al., 2007)—that are used in deep neural networks provide *implicit* regularization, having effects that are not as transparent. In this work, we study another such heuristic—*data augmentation* (Bengio et al., 2011; Dao et al., 2018; Rajput et al., 2019; Chen et al., 2019)—and show that it has a regularizing effect reducing what we call the *rugosity*, or "roughness", of the function learned by the deep network.

Our rugosity measure quantifies how far the mapping $f$ is from a locally linear mapping on the data. Building on the concept of *Hessian eigenmaps* introduced by Donoho & Grimes (2003), we evaluate this deviation from a locally linear mapping using the *tangent Hessian*, assuming the data points $\mathbf{x} \in \mathbb{R}^D$ lie on a low-dimensional manifold $\mathcal{M}$ with dimension $d \ll D$. Intuitively, we would expect functions that generalize well to be smooth—that is, to not have a high amount of rugosity. For example, in classification problems, if we were to partition the feature space by class, then we might expect the set corrresponding to any given class to consist of the union of a few contiguous regions with fairly smooth boundaries. A classification function with high rugosity, particularly near the true class boundaries, will likely have an unstable decision boundary, especially for noisy observations off of the data manifold. See Section 5 for a further discussion of the caveats of using rugosity alone as a rule for learning predictors.

A key part of our analysis leverages a new perspective on deep networks provided by Balestriero & Baraniuk (2018a;b). Let $f(\mathbf{x})$ represent the mapping from the input to output of a deep network constructed using continuous, piecewise affine activations (e.g., ReLU, leaky ReLU, absolute value). Such a network partitions the input space $\mathbb{R}^D$ based on the activation pattern of the network's units (neurons). We call these partitions $\mathscr{Q}_j \subset \mathbb{R}^D$, the *vector quantization (VQ) regions* of the network, and for $\mathbf{x} \in \mathbb{R}^D$ we let $\mathscr{Q}(\mathbf{x})$ denote the VQ region to which $\mathbf{x}$ belongs. Inside one VQ region, $f$ is simply an affine mapping. Consequently, $f$ can be written as a continuous, piecewise affine operator

$$f(\mathbf{x}) = \mathbf{A}_{\mathscr{Q}(\mathbf{x})}\mathbf{x} + \mathbf{b}_{\mathscr{Q}(\mathbf{x})} = \mathbf{A}[\mathbf{x}]\,\mathbf{x} + \mathbf{b}[\mathbf{x}]. \tag{1}$$

For the sake of brevity, we will use the simplified notation $f(\mathbf{x}) = \mathbf{A}[\mathbf{x}]\,\mathbf{x} + \mathbf{b}[\mathbf{x}]$ in the sequel.

This framework enables us to cleanly reason about our extension of the rugosity measure to piecewise affine functions using finite differences, enabling us to make the connection between data augmentation the measure of rugosity via the $\mathbf{A}$ matrices.

**Contributions.** We summarize our contributions as follows: **[C1]** We develop a new *rugosity* measure of function complexity that measures how much a function deviates from locally linear, with a nontrivial extension of this measure to piecewise affine functions based on finite differences. **[C2]** We formally connect the popular heuristic of *data augmentation* to an *explicit rugosity regularization*. **[C3]** We verify experimentally that data augmentation reduces rugosity. **[C4]** We experimentally assess the effect of explicit rugosity regularization in training deep networks.

**Related work.** Recently, Savarese et al. (2019) have shown that, for one-dimensional, one-hidden-layer ReLU networks, an $\ell_2$ penalty on the weights (Tikhonov regularization) is equivalent to a penalty on the integral of the absolute value of the second derivative of the network output. Moreover, several recent works (Belkin et al., 2018; 2019; Hastie et al., 2019) have shown that in some linear and nonlinear inference problems, properly regularized overparameterized models can generalize well. This again indicates the importance of complexity regularization for understanding generalization in deep networks. Rifai et al. (2011) proposed an auto-encoder whose regularization penalizes the change in the encoder Jacobian. Also, Dao et al. (2018) studied the effects of data augmentation by modeling data augmentation as a kernel.

## 2 DATA AUGMENTATION

*Data augmentation* (Wong et al., 2016; Perez & Wang, 2017) is an oft-used yet poorly understood heuristic applied in learning the parameters of deep networks. The data augmentation procedure augments the set of $n$ training data points $\{(\mathbf{x}_i, y_i)\}_{i=1}^n$ to $(m+1)n$ training data points $\left\{\{(\mathbf{x}_i + \mathbf{u}_{ij}, y_i)\}_{j=0}^m\right\}_{i=1}^n$ by applying transformations to the training data $\mathbf{x}_i$ such that they continue to lie on the data manifold $\mathcal{M}$. Example transformations applied to images include translation, rotation, color changes, etc.[1] In such cases, $\mathbf{u}_{ij}$ is the vector difference $\mathbf{x}_j - \mathbf{x}_i$, where $\mathbf{x}_j$ is the translated/rotated image, and $\mathbf{u}_{i0} = \mathbf{0}$. Consider training a deep network with continuous, piecewise affine activations given the original training dataset $\{(\mathbf{x}_i, y_i)\}_{i=1}^n$ by minimizing the loss

$$\mathcal{L} \triangleq \sum_{i=1}^n \ell_i = \sum_{i=1}^n \ell(f(\mathbf{x}_i), y_i), \tag{2}$$

where $\ell(\cdot, \cdot)$ is any convex loss function. After data augmentation, the loss can be written as

$$\mathcal{L}^{\text{aug}} = \frac{1}{m+1}\sum_{i=1}^n \ell_i^{\text{aug}}, \qquad \ell_i^{\text{aug}} \triangleq \ell(f(\mathbf{x}_i), y_i) + \sum_{j=1}^m \ell\left(f\left(\mathbf{x}_i + \mathbf{u}_{ij}\right), y_i\right). \tag{3}$$

In the following result, we obtain an upper bound on the augmented loss, which consists of the sum of the non-augmented loss plus a few terms pertaining to the local similarity of the output function around the training data.

---

[1]Common transformations also include flipping images, which is not a continuous operation; however, our analysis here pertains primarily to continuous transformations, for which the vector differences $\mathbf{u}_{ij}$ are small.

**Theorem 1.** *Consider a deep network with continuous, piecewise affine activations and thus prediction function $f$ as in (1). Assume that $\|\mathbf{x}_i\|_2 \leq R$, and that $f$ has Lipschitz constant $K_1$; that is, $\|\mathbf{A}[\mathbf{x}]\|_2 \leq K_1$, for all $\mathbf{x}$. Further, assume that the loss function $\ell$ has Lipschitz constant $K_2$. Then, for $\|\mathbf{u}_{ij}\|_2 \leq \varepsilon$, $\mathcal{L}^{\mathrm{aug}}$ in (3) can be upper bounded by*

$$
\begin{aligned}
\mathcal{L}^{\mathrm{aug}} \leq \mathcal{L} &+ \frac{RK_2}{m+1} \sum_{i=1}^{n} \sum_{j=1}^{m} \|\mathbf{A}[\mathbf{x}_i + \mathbf{u}_{ij}] - \mathbf{A}[\mathbf{x}_i]\|_2 \\
&+ \frac{K_2}{m+1} \sum_{i=1}^{n} \sum_{j=1}^{m} |b[\mathbf{x}_i + \mathbf{u}_{ij}] - b[\mathbf{x}_i]| + \frac{K_1 K_2 mn\varepsilon}{m+1} + o(\varepsilon).
\end{aligned}
\tag{4}
$$

We focus our attention on the second term of the right-hand side of the above inequality. Denote this term by

$$
\widehat{C}(f) \triangleq \sum_{i=1}^{n} \sum_{j=1}^{m} \|\mathbf{A}[\mathbf{x}_i + \mathbf{u}_{ij}] - \mathbf{A}[\mathbf{x}_i]\|_2.
\tag{5}
$$

In the sequel, we show that this term is effectively a Monte Carlo approximation to a Hessian-based measure of *rugosity* evaluated at the training data points.

## 3 A HESSIAN-BASED RUGOSITY MEASURE

To arrive at $\widehat{C}(f)$ and link data augmentation to an explicit regularization penalty, we first define our new Hessian-based rugosity measure for deep networks with continuous activations. Then, we extend this definition to networks with piecewise-linear activations, and finally we describe its Monte Carlo approximation. We note that while deep networks are our focus, this measure of rugosity is more broadly applicable.

### 3.1 NETWORK WITH SMOOTH ACTIVATIONS

Let $f : \mathbb{R}^D \to \mathbb{R}$ be the prediction function of a deep network whose nonlinear activation functions are smooth functions (e.g., sigmoid). For regression, we can take $f$ as the mapping from the input to the output of the network. For classification, we can take $f$ as the mapping from the input to one of the inputs of the final softmax operation. We assume that the training data points $\{\mathbf{x}_i\}_{i=1}^{n}$ lie close to a $d$-dimensional smooth manifold $\mathcal{M} \subseteq \mathbb{R}^D$. This assumption has been studied in an extensive literature on unsupervised learning (Tenenbaum et al., 2000; Belkin & Niyogi, 2003; Donoho & Grimes, 2003), and holds at least approximately for many practical datasets, including real world images.

For $p \geq 1$, we define the following measure of rugosity:

$$
C_p(f) \triangleq \left( \int_{\mathcal{M}} \left\| \nabla^2_{(\tan)} f(\mathbf{x}) \right\|_F^p \mathrm{d}\mathbf{x} \right)^{1/p}
\tag{6}
$$

where $\nabla^2_{(\tan)} f(\mathbf{x})$ is the Hessian of $f$ at $\mathbf{x}$ in the coordinates of $d$-dimensional affine space tangent to manifold at $\mathbf{x} \in \mathcal{M}$. This is an extension of the quadratic form from the *Hessian eigenmaps* approach of Donoho & Grimes (2003), which is equal to $C_2^2(f)$.

From Donoho & Grimes (2003), we know that $C_p(f)$ measures the average "curviness" of $f$ over the manifold $\mathcal{M}$ and that $C_p(f)$ is zero if and only if $f$ is an affine function on $\mathcal{M}$. In the simplistic case of one-dimensional data and one-hidden-layer networks, Savarese et al. (2019) have related $C_1(f)$ to the sum of the squared Frobenius norms of the weight matrices.

### 3.2 NETWORK WITH CONTINUOUS PIECEWISE AFFINE ACTIVATIONS

We are primarily interested in deep networks constructed using piecewise affine activations (e.g., ReLU, leaky ReLU, absolute value). Since $f$ is now piecewise affine and thus not continuously differentiable, the Hessian is not well defined in its usual form. However, using the formulation (1)

we can extend our rugosity measure to this case. Let $\varepsilon > 0$. For $\mathbf{x}$ not on the boundary of a VQ partition and $\mathbf{u}$ an arbitrary unit vector, we define

$$\mathbf{H}_\varepsilon(\mathbf{x})[\mathbf{u}] \triangleq \frac{1}{\varepsilon} \left( \mathbf{A}[\mathbf{x} + \varepsilon\mathbf{u}] - \mathbf{A}[\mathbf{x}] \right) \tag{7}$$

if $\mathbf{u}$ is on the $d$-dimensional affine space tangent to the data manifold $\mathcal{M}$ at $\mathbf{x}$ and $\mathbf{H}_\varepsilon(\mathbf{x})[\mathbf{u}] = \mathbf{0}$, otherwise. Note that $\mathbf{A}[\mathbf{x}]$ is a (weak) gradient of $f$ at $\mathbf{x}$, and therefore this definition agrees with the finite element definition of the Hessian. For smooth $f$ and $\varepsilon \to 0$, this recovers the Hessian (Milne-Thomson, 2000; Jordan & Jordán, 1965). Thus, for a network with continuous, piecewise affine activations, we define

$$C_{p,\varepsilon}(f) \triangleq \sqrt{d} \left( \int_{\mathcal{M}} \left( \mathbb{E}_\mathbf{u} \left\| \mathbf{H}_\varepsilon(\mathbf{x})[\mathbf{u}] \right\|_2^2 \right)^{p/2} d\mathbf{x} \right)^{1/p} \tag{8}$$

where $\mathbf{u}$ is uniform over the unit sphere. Comparing with (6), this definition is consistent with the definition of the distributional derivative for piecewise constant functions and can be seen as measuring the changes in the local slopes of the piecewise affine spline realized by the network.

### 3.3 MONTE CARLO APPROXIMATION

We can estimate $C_p(f)$ using a Monte Carlo approximation $\widetilde{C}_p(f)$ based on the training data $\mathbf{x}_i$. When the network has *smooth activations*, we have

$$\widetilde{C}_p(f) \triangleq \left( \frac{1}{n} \sum_{i=1}^{n} \left\| \nabla^2_{(\text{tan})} f(\mathbf{x}_i) \right\|_F^p \right)^{1/p} . \tag{9}$$

We can also apply the Monte Carlo method to estimate $\|\nabla^2_{(\text{tan})} f(\mathbf{x}_i)\|_F^2$. If $\mathbf{u} \in \mathbb{R}^d$ is chosen uniformly at random on the unit sphere, then for $\mathbf{A} \in \mathbb{R}^{d \times d}$ we have

$$\mathbb{E} \|\mathbf{A}\mathbf{u}\|_2^2 = \mathbb{E} \sum_{i,j} u_i u_j (\mathbf{A}^T \mathbf{A})_{ij} = \frac{1}{d} \|\mathbf{A}\|_F^2 . \tag{10}$$

For smooth $f$, we have

$$\nabla^2 f(\mathbf{x})\mathbf{u} = \lim_{\delta \to 0} \frac{1}{\delta} \left[ \nabla f(\mathbf{x} + \delta\mathbf{u}) - \nabla f(\mathbf{x}) \right] . \tag{11}$$

Therefore, choosing $\mathbf{u}_j$ uniformly at random with $\|\mathbf{u}_j\|_2 = 1$ on the $d$-dimensional subspace tangent to the manifold at $\mathbf{x}$, plugging (10), (11) into (9) yields the following approximation for $C_p(f)$ for small $\delta > 0$

$$\widetilde{C}_{p,\delta}(f) = \left( \frac{d^{p/2}}{n \delta^p m^{p/2}} \sum_{i=1}^{n} \left( \sum_{j=1}^{m} \left\| \nabla f(\mathbf{x}_i + \delta\mathbf{u}_j) - \nabla f(\mathbf{x}_i) \right\|_2^2 \right)^{p/2} \right)^{1/p} . \tag{12}$$

When the network has continuous, piecewise affine activations, Monte Carlo approximation of $C_{p,\varepsilon}(f)$ based on the training data $\mathbf{x}_i$ yields

$$\widetilde{C}_{p,\varepsilon}(f) \triangleq \left( \frac{d^{p/2}}{n} \sum_{i=1}^{n} \left( \mathbb{E}_\mathbf{u} \left\| \mathbf{H}_\varepsilon(\mathbf{x}_i)[\mathbf{u}] \right\|_2^2 \right)^{p/2} \right)^{1/p} . \tag{13}$$

For such a network, using (7) yields

$$\widetilde{C}_{p,\varepsilon}(f) = \left( \frac{d^{p/2}}{n \varepsilon^p m^{p/2}} \sum_{i=1}^{n} \left( \sum_{j=1}^{m} \left\| \mathbf{A}[\mathbf{x}_i + \varepsilon\mathbf{u}_j] - \mathbf{A}[\mathbf{x}_i] \right\|_2^2 \right)^{p/2} \right)^{1/p} . \tag{14}$$

as our approximation.

Table 1: Data augmentation reduces rugosity in a classification task with a range of deep networks and datasets. We tabulate the converged values for $\widetilde{C}_2^2$ and $J$ on the training and test data without and with data augmentation (denoted by DA). Missing values are omitted due to time constraints.

| | $\widetilde{C}_{2,\text{train}}^2$ | $\widetilde{C}_{2,\text{test}}^2$ | $J_{\text{train}}$ | $J_{\text{test}}$ | Test accuracy (%) |
|---|---|---|---|---|---|
| CNN (MNIST) | 1.57e-09 | 0.036 | 33.0 | 33.2 | 99.6 |
| CNN+DA (MNIST) | 1.56e-09 | 0.074 | 6.79 | 6.83 | 99.6 |
| CNN (SVHN) | 8.3e-10 | 0.022 | 67.0 | 63.3 | 95.6 |
| CNN+DA (SVHN) | 6.32e-10 | 0.019 | 108.5 | 103.2 | 95.5 |
| CNN (CIFAR10) | 1.54e-09 | 0.10 | 102.0 | 102.0 | 87.4 |
| CNN+DA (CIFAR10) | 2.58e-09 | 0.11 | 137.0 | 138.0 | 91.7 |
| CNN (CIFAR100) | 6.3e-08 | 1.51 | — | — | 61.1 |
| CNN+DA (CIFAR100) | 7.0e-08 | 1.40 | — | — | 68.3 |
| ResNet (MNIST) | 0.086 | 0.20 | 23.7 | 23.7 | 99.4 |
| ResNet+DA (MNIST) | 0.021 | 0.016 | 17.7 | 17.8 | 99.4 |
| Large ResNet (MNIST) | 0.061 | 0.079 | — | — | 99.5 |
| Large ResNet+DA (MNIST) | 0.019 | 0.025 | — | — | 99.5 |
| ResNet (SVHN) | 0.08 | 0.09 | 48.3 | 47.8 | 93.9 |
| ResNet+DA (SVHN) | 0.01 | 0.01 | 56.5 | 56.5 | 94.1 |
| ResNet (CIFAR10) | 0.43 | 0.50 | 105.0 | 105.4 | 84.9 |
| ResNet+DA (CIFAR10) | 0.10 | 0.12 | 107.3 | 107.8 | 91.0 |
| ResNet (CIFAR100) | 4.32 | 4.39 | — | — | 49.3 |
| ResNet+DA (CIFAR100) | 1.08 | 1.14 | — | — | 64.5 |

## 3.4 CONNECTION TO DATA AUGMENTATION

We can now identify the connection between the rugosity measure in (14) and data augmentation. Note that for $p = 1$, the quantity in (14) is very similar (up to normalization) to $\widehat{C}(f)$ from Theorem 1. In fact, we obtain $\widehat{C}(f)$ by replacing $L^2$ norm with respect to the measure $dP(\mathbf{u})$ in (13) with the $L^1$ norm. We also note that in data augmentation methods that generate data by continuous deviations from the original image, such as translation, rotation, and color change, the vectors $\mathbf{u}_{ij}$ in (5) are such that $\mathbf{x}_i + \mathbf{u}_{ij} \in \mathcal{M}$. Thus, a sufficiently rich set of small augmentations yields a distribution on $\mathbf{u}_{ij}$ with very similar support to the uniform distribution over unit vectors on the subspace tangent to the manifold.

The above fact suggests that using Hessian-based rugosity measure as an explicit regularization term in training a neural network would improve the prediction accuracy of a deep network by having a similar effect as data augmentation through minimizing an upper bound on the augmented loss $\mathcal{L}^{\text{aug}}$. Remarkably, the converse is observed in our simulations. Using data augmentation in training deep networks decreases the rugosity of the prediction function.

## 4 EXPERIMENTS

We now explore, through a range of experiments, the connections between data augmentation and explicit rugosity regularization. First, we show that networks trained using data augmentation feature a significant decrease in rugosity that accompanies an increase in test accuracy. Second, we show that rugosity can be employed as an explicit regularization penalty in training sufficiently overparameterized networks at no cost to training accuracy. Although using this explicit regularization decreases the rugosity of the prediction function of the deep network, surprisingly, it does *not* yield an increase in test accuracy, in general. We discuss this observation further in Section 5.

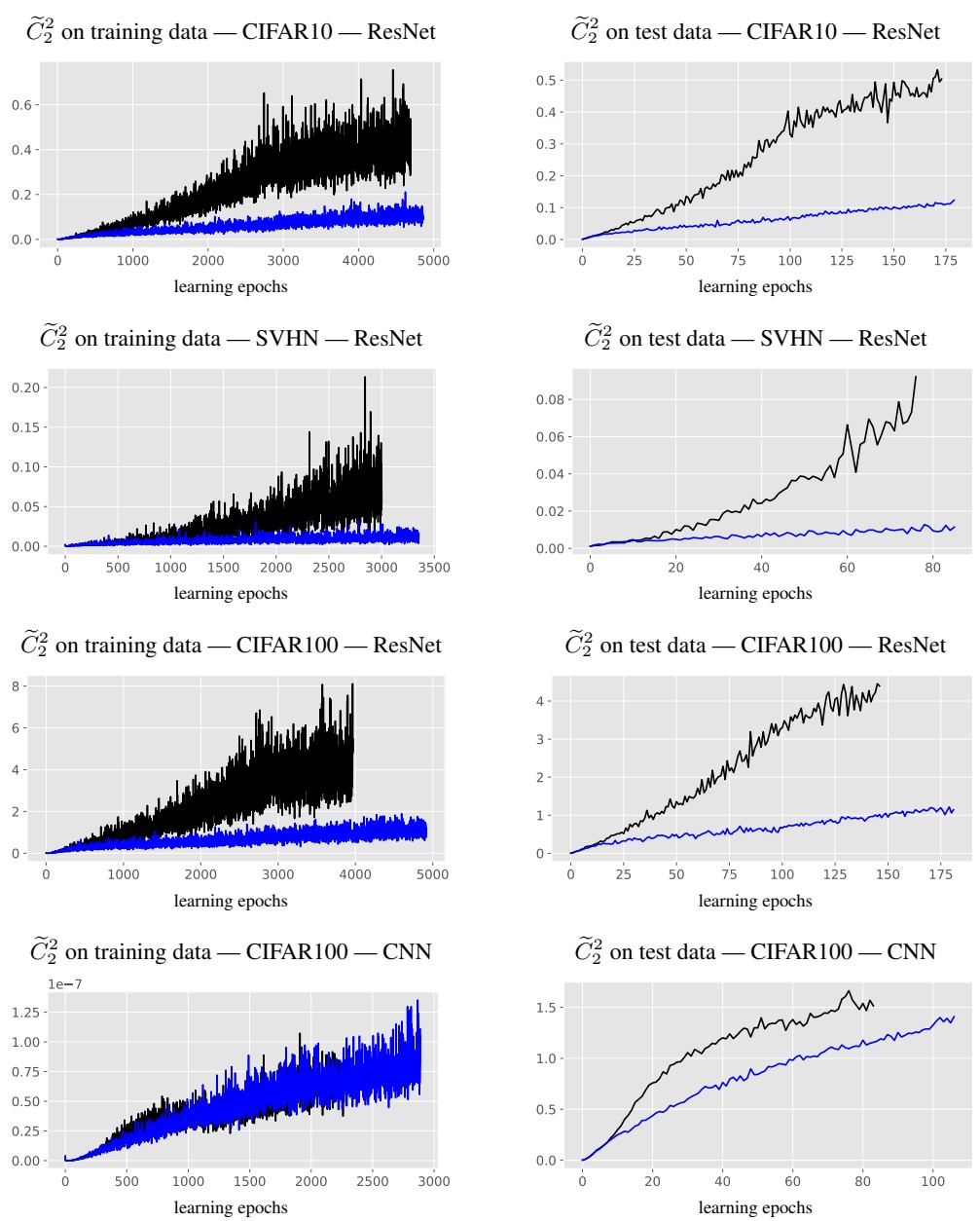

Figure 1: Data augmentation reduces rugosity. The blue and black curves correspond to experiments with and without data augmentation, respectively.

## 4.1 DATA AUGMENTATION REDUCES RUGOSITY

In Table 1, we tabulate the rugosity $\widetilde{C}_{2,\varepsilon}^2(f)$ as well as the *Jacobian norm* from Novak et al. (2018), a similar proposed complexity measure, computed as

$$J(f) = \frac{1}{n} \sum_{i=1}^{n} \|\mathbf{A}[\mathbf{x}_i]\|_F, \tag{15}$$

for residual network (ResNet) and convolutional neural network (CNN) architectures (described in Appendix A.2) on several image classification tasks, trained with and without data augmentation. The data augmentation used in this experiment is translation vertically and horizontally each by no more than 4 pixels, as well as flipping horizontally. In Figure 1, we plot the rugosity over training epochs for a subset of these architecture and dataset combinations.

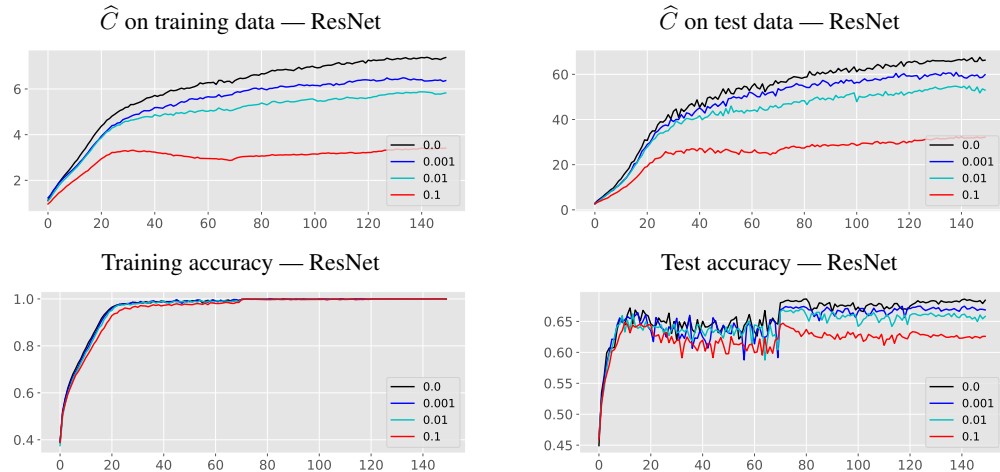

Figure 2: Explicit rugosity regularization. We plot the rugosity and accuracy on the training and test data over training time for several different values of the penalty weighting factor $\lambda$. We observe that rugosity can be used as an explicit regularization penalty at no cost to training accuracy but there are no test accuracy gains.

We can make several observations from the results in Table 1. First, for the ResNet architecture, there is a clear reduction in rugosity when using data augmentation, while there is no such decrease in Jacobian norm (in fact, we sometimes observe an *increase* for both network types). We also observe that the CNN results in smaller $\widetilde{C}_2$ than the ResNet on both the training and test data. In fact, we observe that $\widetilde{C}_{2,\text{train}}$ is almost zero for CNNs on all four datasets. This suggests that the prediction function is almost linear within $\varepsilon$ of the training data. However, the rugosity is significantly higher when measured using the test data. We also note based on Figure 1 that while the CNN does not show much reduced rugosity at the end of training, there is a significant gap between the test rugosities with and without data augmentation *during* training. These peculiarities of purely convolutional networks are interesting properties that beg investigation in future work. Lastly, as one might expect, training the same network for a more complex task (e.g., classification with CIFAR100 vs. CIFAR10) results in higher rugosity in the learned network.

### 4.2 RUGOSITY AS AN EXPLICIT REGULARIZATION PENALTY

In Figure 2, we plot the rugosity as well as the training and test accuracies when training a network with and without using rugosity as an explicit regularization term. Specifically, we optimize the regularized loss $\mathcal{L}+\lambda\widehat{C}(f)$, where $\widehat{C}(f)$ is computed using using vertical and horizontal translations of up to 2 pixels. As we observe in these plots, the network can achieve almost zero training error even after adding the extra penalty term. Also, as expected, adding the regularization term decreases the rugosity of the obtained prediction function. However, somewhat surprisingly, minimizing the rugosity-penalized loss does not necessarily result in improved test accuracy. In fact, increasing the weight $\lambda$ on the regularization term results in lower test accuracy even though the resulting network has zero training error. We discuss a few possible reasons for this phenomenon in Section 5. This observation points to the difference between data augmentation and explicit rugosity regularization. Although using data augmentation reduces rugosity, it also provides an improvement in test accuracy not achieved using rugosity alone as an explicit regularization.

## 5 DISCUSSION

Now that we have made the connection between data augmentation and rugosity regularization, the question remains of whether an explicit rugosity penalty in the optimization criterion is of value in its own right. Arguing in favor of such an approach, one may compare our Hessian-based rugosity measure with the notion of counting the "number of VQ regions" (Hanin & Rolnick, 2019) in a

ReLU network. One might argue that the rugosity measure provides a more useful quantification of the network output complexity, because it explicitly takes into account the changes in the output function across the VQ regions. For instance, consider the analysis of infinitely wide networks, which have been used to help understand the convergence properties and performance of deep networks (Lee et al., 2017; Mei et al., 2019; Arora et al., 2019). The number of VQ regions can be infinite in such networks; however, the rugosity measure can remain bounded.

For instance, consider the *minimum-rugosity interpolator*

$$f^* \triangleq \arg\min_f C_{p,\varepsilon}(f) \text{ s.t. } g(f(\mathbf{x}_i)) = y_i \ \forall i, \tag{16}$$

where $g(\cdot)$ is the identity mapping in regression problems and the $\arg\max(\cdot)$ function in classification problems. Here, $f^*$ is the least rugous function that interpolate the training data.

In the classification setting, observe that, for any $\alpha > 0$, $g(f(\mathbf{x})) = g(\alpha f(\mathbf{x}))$. Thus, for any function $f$, we can make $C_p(f)$ and $C_{p,\varepsilon}(f)$ arbitrarily small by rescaling $f$ by an arbitrarily small $\alpha$ while still predicting the same labels for the training data. This is also true of other proposed complexity measures based on derivatives such as the Jacobian norm. In the binary classification setting, where $f$ can have unidimensional output and $g(\cdot) = \text{sgn}(\cdot)$, the effect is that $f$ need only rise slightly above zero to predict the correct label for positive training examples and fall slightly below zero for negative training examples. Suppose that such a function were to perfectly classify all points on the training data manifold, but consider a test data point $\mathbf{x} + \mathbf{u}$, where $\mathbf{x}$ lies on the training data manifold $\mathcal{M}$ but $\mathbf{u}$ is some noise orthogonal to the tangent space of the manifold at $\mathbf{x}$. Then by Taylor's theorem, $f(\mathbf{x} + \mathbf{u}) = f(\mathbf{x}) + \nabla_{\mathbf{z}} f(\mathbf{z})^\top \mathbf{u}$ for $\mathbf{z} = \mathbf{x} + t\mathbf{u}$ for some $t \in [0, 1]$. For a function $f$ learned by a deep network, it is unlikely that $\nabla_{\mathbf{z}} f(\mathbf{z})$ will be exactly equal to zero, effectively changing the classification function from thresholding $f(\mathbf{x})$ at 0 to thresholding $f(\mathbf{x})$ at some small positive or negative value. For $f$ that has been scaled to have very small output values, the result is that classification performance on noisy data will be poor.

The regression setting is not immune to problems. Savarese et al. (2019, Claim 5.2) showed the existence of a class of regression functions realizable by infinite-width deep networks that perfectly interpolate any finite set of training points. Functions in this class can have arbitrarily small Hessian norm integral—that is, $C_1(f)$—and have a constant value almost everywhere, therefore generalizing very poorly. However, for sufficiently large $p$ (dependent on $d$), this particular argument does not hold for the more general $C_p(f)$. Further, realizable networks are of finite width and are heavily regularized in training, and so it is not clear whether such functions can be learned in practice. More thoroughly quantifying the shortcomings of using $C_p$ to reason about deep networks as well as developing more useful measures of network complexity remains an exciting area for future work.

ACKNOWLEDGMENTS

We thank [redacted] for helpful discussions on the rugosity measure in an anonymous personal communication.

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

# A APPENDIX

## A.1 PROOF OF THEOREM 1

Let $\ell_{ij} \triangleq \ell(f(\mathbf{x}_i + \mathbf{u}_{ij}), y_i)$. Using (1), we have

$$\ell_{ij} = \ell\big(\mathbf{A}[\mathbf{x}_i + \mathbf{u}_{ij}](\mathbf{x}_i + \mathbf{u}_{ij}) + b[\mathbf{x}_i + \mathbf{u}_{ij}], y_i\big) \tag{17}$$

$$= \ell\big(\mathbf{A}[\mathbf{x}_i]\mathbf{x}_i + b[\mathbf{x}_i] + (\mathbf{A}[\mathbf{x}_i + \mathbf{u}_{ij}] - \mathbf{A}[\mathbf{x}_i])\mathbf{x}_i + (b[\mathbf{x}_i + \mathbf{u}_{ij}] - b[\mathbf{x}_i]) + \mathbf{A}[\mathbf{x}_i + \mathbf{u}_{ij}]\mathbf{u}_{ij}, y_i\big). \tag{18}$$

Therefore, for $\tilde{\ell}_{ij}$, the first order approximation of $\ell_{ij}$ around $\ell_i = \ell(f(\mathbf{x}_i), y_i)$ we obtain

$$\tilde{\ell}_{ij} = \ell_i + \big[(\mathbf{A}[\mathbf{x}_i + \mathbf{u}_{ij}] - \mathbf{A}[\mathbf{x}_i])\mathbf{x}_i + (b[\mathbf{x}_i + \mathbf{u}_{ij}] - b[\mathbf{x}_i]) + \mathbf{A}[\mathbf{x}_i + \mathbf{u}_{ij}]\mathbf{u}_{ij}\big]^\top \nabla_{f(\mathbf{x}_i)} \ell(f(\mathbf{x}_i), y_i). \tag{19}$$

Under the conditions of the theorem, we obtain

$$\tilde{\ell}_{ij} \le \ell_i + RK_2 \left\|\mathbf{A}[\mathbf{x}_i + \mathbf{u}_{ij}] - \mathbf{A}[\mathbf{x}_i]\right\|_2 + K_2 \left|b[\mathbf{x}_i + \mathbf{u}_{ij}] - b[\mathbf{x}_i]\right| + \varepsilon K_1 K_2. \tag{20}$$

Summing up over $\mathbf{x}_i$ and $\mathbf{u}_{ij}$ yields the following bound for $\widetilde{\mathcal{L}}^{\mathrm{aug}}$, the first order approximation of $\mathcal{L}^{\mathrm{aug}}$ for small $\varepsilon$,

$$\widetilde{\mathcal{L}}^{\mathrm{aug}} = \frac{1}{m+1} \sum_{i=1}^{n} \left(\ell_i + \sum_{j=1}^{m} \tilde{\ell}_{ij}\right) \tag{21}$$

$$\le \mathcal{L} + \frac{RK_2}{m+1} \sum_{i=1}^{n} \sum_{j=1}^{m} \left\|\mathbf{A}[\mathbf{x}_i + \mathbf{u}_{ij}] - \mathbf{A}[\mathbf{x}_i]\right\|_2$$

$$+ \frac{K_2}{m+1} \sum_{i=1}^{n} \sum_{j=1}^{m} \left|b[\mathbf{x}_i + \mathbf{u}_{ij}] - b[\mathbf{x}_i]\right| + \frac{K_1 K_2 mn\varepsilon}{m+1} \tag{22}$$

which completes the proof.

## A.2 EXPERIMENTAL DETAILS

The experiments in Figure 1 and Table 1 used the following parameters: batch size of 16, Adam optimizer with learning scheduled at 0.005 (initial), 0.0015 (epoch 100) and 0.001 (epoch 150). The default training/test split was used for all datasets. The validation set consists of 15% of the training set sampled randomly.

For the experiments in Figure 2, the number of training samples is $10,000$ sampled randomly from the training set. The test set still consists of the CIFAR10 test set of $10,000$ samples. The learning rate is 0.0001 and is divided by 3 at 75 epochs and again at 120 epochs. The images are renormalized to have zero mean and infinity norm of one.

### A.2.1 CNN ARCHITECTURE

```
Conv2D(Number Filters=96, size=3x3, Leakiness=0.01))

Conv2D(Number Filters=96, size=3x3, Leakiness=0.01))

Conv2D(Number Filters=96, size=3x3, Leakiness=0.01))

Pool2D(2x2)

Conv2D(Number Filters=192, size=3x3, Leakiness=0.01))

Conv2D(Number Filters=192, size=3x3, Leakiness=0.01))
```

```
Conv2D(Number Filters=192, size=3x3, Leakiness=0.01))

Pool2D(2x2)

Conv2D(Number Filters=192, size=3x3, Leakiness=0.01))

Conv2D(Number Filters=192, size=1x1, Leakiness=0.01))

Conv2D(Number Filters=Number Classes, size=1x1, Leakiness=0.01))

GlobalPool2D(pool_type='AVG'))
```

### A.2.2 RESNET AND LARGE RESNET ARCHITECTURES

The ResNets follow the original architecture (Zagoruyko & Komodakis, 2016) with depth 2, width 1 for the ResNet and depth 4, width 2 for the Large ResNet.

### A.2.3 IMPLEMENTATION NOTES

In a ReLU network with $L$ layers, for a given $\mathbf{x}$, $\mathbf{A}[\mathbf{x}]$ can be computed via

$$\mathbf{A}[\mathbf{x}] = \mathbf{W}^{(L)} \left( \prod_{\ell=1}^{L} \mathbf{D}^{(L-\ell)}[\mathbf{x}] \mathbf{W}^{(L-\ell)} \right), \tag{23}$$

where $\mathbf{W}^{(\ell)}$ is the weight matrix of the $\ell$-th layer and $\mathbf{D}^{(\ell)}[\mathbf{x}]$ is a diagonal matrix with $\left(\mathbf{D}^{(\ell)}[\mathbf{x}]\right)_{rr} = 1$ if the output of the $r$-th ReLU unit in the $\ell$-th layer is nonzero (when the unit is "active") with $\mathbf{x}$ as input and $\left(\mathbf{D}^{(\ell)}[\mathbf{x}]\right)_{rr} = 0$ if the ReLU ouput is zero. This enables us to use $\widehat{C}(f)$ as a regularization penalty in training real networks.

