# OpenReview forum: "Implicit Rugosity Regularization via Data Augmentation"
_ICLR.cc/2020/Conference — Reject_

### Official Review · AnonReviewer1 · 2019-10-22
**Official Blind Review #1**

**Rating:** 3

**Review:**

This paper shows (theorem 1) that data augmentation (DA) induces a reduction of rugsity on the loss function associated to the model. Here rugosity is defined as a measure of the curvature (2nd order) of the function. However, the two concepts seems to be different because the authors empirically show that directly reducing the rugosity of a network does not improve generalization (in contrast to DA).

I lean to reject this paper because the contributions, even if interesting, do not lead to any new understanding of the topic. More in detail, data augmentation improves the generalization on deep learning models. This paper shows that DA induces rugosity (theorem 1), but rugosity does not improve generalization (empirically). Thus, rugosity is not responsible for generalization, which is the interesting property that we care about.

The paper is well written and easy to follow, however I found the actual contribution limited because:
- The definition of rugosity is an extension of (Donoho & Grimes (2003)) in which the extension is not really improving anything or used anywhere in the paper.
- The Hessian-based rugosity analysis of DA is correct, but it does not help to understand the generalization performance or any other useful property of DA.

Additional Comments:
- In 3.4 second paragraph the authors suggest that reducing rugosity can improve generalization as DA, but later we see that this is not the case.
- The entire paper seems written with the idea of using rugosity as a surrogate of DA, but at the end it does not work


**Experience Assessment:**

I have published one or two papers in this area.

**Review Assessment: Checking Correctness Of Derivations And Theory:**

I assessed the sensibility of the derivations and theory.

**Review Assessment: Checking Correctness Of Experiments:**

I assessed the sensibility of the experiments.

**Review Assessment: Thoroughness In Paper Reading:**

I read the paper thoroughly.

---

> ### Author Response · Authors · 2019-11-13
> **Response to Reviewer 1**
>
> We thank the reviewer for positive comments and useful suggestions. We provide the following responses for the comments.
>
> 1> The definition of rugosity is an extension of (Donoho & Grimes (2003)) in which the extension is not really improving anything or used anywhere in the paper.
>
> The rugosity measure that we defined and used is in fact inspired by the tangent Hessian measure proposed by (Donoho & Grimes (2003)). We have modified the tangent Hessian integral measure provided in this paper to make it suitable to use for piecewise linear functions and for making it easier to compute. We believe that this measure can provide useful information about the landscape and complexity of the prediction function generated by the deep network.
>
> 2> Data augmentation improves the generalization on deep learning models. This paper shows that DA induces rugosity (theorem 1), but rugosity does not improve generalization (empirically). Thus, rugosity is not responsible for generalization, which is the interesting property that we care about. The Hessian-based rugosity analysis of DA is correct, but it does not help to understand the generalization performance or any other useful property of DA.
>
> We agree that our empirical results do not show significant classification performance improvement as a result of using rugosity as explicit regularization. However, the effect of data augmentation on rugosity (and also generalization) is a significant observation and points to the question of what other properties data augmentation possesses that lead to improve generalization. We believe that this is an important question that requires a more thorough and comprehensive understanding of regularization in the overparameterized (interpolating) regime. What our results suggest is that, although there is a close connection between rugosity and data augmentation, rugosity (or smoothness) cannot by itself explain the entire effect of data augmentation on generalization.
>
> In addition, understanding generalization is not the primary goal of our paper. We believe that our rugosity measure can be a very useful data-driven measure for understanding the prediction function generated by a deep network. One of our applications was understanding data augmentation, which we illustrated through theoretical and empirical analysis. We showed the close connection between data augmentation and rugosity which suggests that rugosity can be a more effective complexity measure than other common complexity measures, e.g., the Jacobian. Further, there can be many other properties of deep networks, such as adversarial robustness, that rugosity can help us to better understand and improve. We believe that our work is only the first step in this direction.
>
> 3> In 3.4 second paragraph the authors suggest that reducing rugosity can improve generalization as DA, but later we see that this is not the case.
>
> The effect of rugosity as explicit regularization on improving generalization is what can be initially expected from Theorem 1 and the empirical results showing the connection between data augmentation and rugosity. However, surprisingly, this is not what we observed in our further experiments. This suggests that understanding generalization requires a more comprehensive understanding of properties (other than just the rugosity or smoothness) of the function generated by a deep network.
>
> 4> The entire paper seems written with the idea of using rugosity as a surrogate of DA, but at the end it does not work.
>
> We do not suggest to use rugosity as a surrogate for data augmentation. Rather, we propose rugosity as a useful, data-driven measure for studying a deep network and the complexity of the prediction function it produces. While we showed that rugosity has a close connection to data augmentation, our experiments show also that rugosity, by itself, cannot completely explain the generalization properties of deep nets or data augmentation.

---

### Official Review · AnonReviewer2 · 2019-10-23
**Official Blind Review #2**

**Rating:** 3

**Review:**

The paper aims to explain the regularization and generalization effects of data augmentation commonly used in training Neural Networks.
It suggests a novel measure of "rugosity" that measures a function's diversion from being locally linear and explores the connection between data augmentation and the decrease in rugosity.
It further suggests the explicit use of rugosity measure as a regularization during training to replace need for data augmentation.
The paper is very well written and both the positive and negative findings are clearly presented and discussed.
Cons:
- The main contribution of the paper, in my view, is the suggestion of using rugosity as a explicit regularization for training Neural Networks. Nevertheless, all the results in the paper show a negative impact of this on the test accuracy which is contradicting to the proposition.
This result has been discussed in section 5 but without much evidence to the explanations mentioned. The connection is very interesting but I believe further work is needed to explain those negative results on test accuracy.
- The difference in finding (Table 1) between the CNN and ResNet networks can be more discussed.
- Additional tasks (like regression) or even toy examples can be useful in further explaining the connection between rugosity and generalization to test data.


**Experience Assessment:**

I do not know much about this area.

**Review Assessment: Checking Correctness Of Derivations And Theory:**

I did not assess the derivations or theory.

**Review Assessment: Checking Correctness Of Experiments:**

I assessed the sensibility of the experiments.

**Review Assessment: Thoroughness In Paper Reading:**

I read the paper at least twice and used my best judgement in assessing the paper.

---

> ### Author Response · Authors · 2019-11-13
> **Response to Reviewer 2**
>
> We thank the reviewer for positive comments and useful suggestions. We provide the following responses for the comments.
>
> 1> The main contribution of the paper, in my view, is the suggestion of using rugosity as a explicit regularization for training Neural Networks. Nevertheless, all the results in the paper show a negative impact of this on the test accuracy which is contradicting to the proposition.
>
> We agree that we did not observe a significant improvement in generalization error as a result of using rugosity as an explicit regularization in our experiments. However, this is not the only contribution of our paper. As we show in the paper, rugosity provides a data-driven complexity measure for the prediction function of the deep network that can help demystify generalization and (implicit) regularization. As an example, we show how rugosity can be used to understand the effects of data augmentation. We plan to  perform further exploration on the effects of using rugosity as an explicit regularization in our revised paper. For example, we expect that using rugosity as explicit regularization can improve adversarial robustness.
>
> 2> The difference in finding (Table 1) between the CNN and ResNet networks can be more discussed. Additional tasks (like regression) or even toy examples can be useful in further explaining the connection between rugosity and generalization to test data.
>
> We appreciate these useful suggestions. We agree that these points require further examination, and we will include this further analysis in our revision.

---

### Official Review · AnonReviewer4 · 2019-11-01
**Official Blind Review #4**

**Rating:** 3

**Review:**

This paper shows that a penalty term called rugosity captures the implicit regularization effect of deep neural networks with ReLU (and piecewise affine in general) activation. Roughly, rugosity measures how far the function parametrized as a deep network deviates from a locally linear function.

The paper starts by showing that the amount of training loss increased from adding data augmentation is upper bounded in terms of (roughly) a Monte Carlo approximate to a Hessian based measure of rugosity. It then formally derives this measure of rugosity for networks with continuous piecewise affine activations. Finally, experimental evaluation for classification tasks on MNIST, SVHN and CIFAR shows that data augmentation indeed reduces the rogusity by a significant amount particularly when using the ResNet structure. A somehow surprising message is, however, that if one imposes explicit regularization with rugosity in lieu of data augmentation, then the better generalization usually seen from data augmentation no longer presents, though one does get a network with smaller rugosity.

Comments:

It is quite interesting to see that the rugosity measure proposed in the paper captures at least some aspects of the implicit regularization effect of data augmentation both in terms of theory (i.e. Theorem 1) and practical observations. My feeling is that rugosity is mostly a measure of the smoothness of the function parametrized by the neural network. From that perspective, how is the rugosity as a smoothness measurement for neural networks with piecewise affine activations different from the Lipschitz constant for general neural networks? My guess is that data augmentation also decreases the Lipschitz constant of a neural network near the training data points, but regardless of whether this is true or not, it is not clear if and how rugosity is better than Lipschitz constant for characterizing the implicit regularization of data augmentation.

In addition, there have been many recent studies on showing that gradient penalty / Lipschitz regularization are useful for achieving better generalization and adversarial robustness, see e.g. [a,b,c]. The results in this paper on showing that regularizing rugosity does not improve accuracy seem to contradict with the conclusion of these prior studies. It is unclear to me whether this is caused by insufficient experimentation or if there is any fundamental difference between rugosity and Lipschitz regularization that I am missing.

[a] Finlay et al., Lipschitz regularized deep neural networks generalize and are adversarially robust
[b] Gouk et al., Regularisation of Neural Networks by Enforcing Lipschitz Continuity
[c] Thanh-Tung et al., Improving generalization and stability of GANs






**Experience Assessment:**

I have read many papers in this area.

**Review Assessment: Checking Correctness Of Derivations And Theory:**

I assessed the sensibility of the derivations and theory.

**Review Assessment: Checking Correctness Of Experiments:**

I assessed the sensibility of the experiments.

**Review Assessment: Thoroughness In Paper Reading:**

I read the paper at least twice and used my best judgement in assessing the paper.

---

> ### Author Response · Authors · 2019-11-13
> **Response to Reviewer 4**
>
> We thank the reviewer for their positive comments and useful suggestions. We provide the following response.
>
> 1> How is the rugosity as a smoothness measurement for neural networks with piecewise affine activations different from the Lipschitz constant for general neural networks?
>
> Two main features of our measure distinguish it from the Lipschitz constant: 1) Rugosity depends on the Hessian of the function generated by the deep network, which is a second-order smoothness measure.
> In contrast, the Lipschitz constant is a first-order measure that depends on the first derivative (Jacobian) of the network. In other words, rugosity quantifies how much the function generated by a deep network differs from an affine mapping over the input space. We believe rugosity provides a better measure for complexity, especially when the network consists of continuous piecewise linear activations that result in a continuous piecewise linear prediction function. 2) In many applications, for example when the input data consists of natural images, the training data points $x_i \in \mathbb{R}^{D}$ lie on a lower-dimensional manifold $M$ of dimension $d \ll D$. We can exploit this local geometrical structure to evaluate the prediction mapping $f$ as a function of the manifold local coordinates and compute the rugosity on the data manifold. This result is a natural data-driven complexity measure for $f$ that evaluates its complexity over the signal space of importance. In contrast, the standard Lipschitz constant considers the entire input space $\mathbb{R}^{D}$, which might be not relevant.
>
> In addition, as we demonstrate through our empirical results in Section 4.1 and Table 1, for classification tasks on the CIFAR10 and SVHN datasets, rugosity better reflects the difference between training with and without data augmentation. Table 1 shows that data augmentation reduces $\widetilde C$ when used for training on these datasets but has no effect (or increases) the Jacobian measure $J$. This suggests that rugosity is a more informative complexity measure than the Jacobian.
>
> 2> There have been many recent studies on showing that gradient penalty / Lipschitz regularization are  useful  for  achieving  better  generalization  and  adversarial robustness. The  results  in this  paper  on  showing  that  regularizing  rugosity  does  not  improve  accuracy  seem  to  contradict  with  the conclusion of these prior studies.  It is unclear to me whether this is caused by insufficient experimentation or if there is any fundamental difference between rugosity and Lipschitz regularization that I am missing.
>
> We thank the reviewer for mentioning these interesting works. We plan to apply our rugosity measure as an explicit regularization penalty in the settings discussed in these papers in order to see the effect on generalization and adversarial robustness in our revised paper. In fact, we suspect that using rugosity as an explicit regularization should improve adversarial robustness, and we will include an empirical study of this hypothesis in our next revision. In addition, we should note that there are results in the literature confirming our observation that using the Jacobian as explicit regularization does not have a significant effect on generalization:  Hoffman et al., ``Robust Learning with Jacobian Regularization.''

---

### Decision · Program_Chairs · 2019-12-19

**Decision:**

Reject

**Comment:**

This paper aims to study the effect of data augmentation of generalization performance. The authors put forth a measure of rugosity or "roughness" based on the tangent Hessian of the function reminiscent of a classic result by Donoho et. al. The authors show that this measure changes in tandem with how much data augmentation helps. The reviewers and I concur that the rugosity measure is interesting. However, as the reviewer mention the main draw back of this paper is that this measure of rugosity when made explicit does not improve generalization. This is the main draw back of the paper. I agree with the authors that this measure is interesting in itself. However, I think in its current form the paper is not ready for prime time and recommend rejection. That said, I believe this paper has a lot of potential and recommend the authors to rewrite and carry out more careful experiments for a future submission.